# Prospective discovery of small molecule enhancers of an E3 ligase-substrate interaction

Kyle R. Simonetta[1], Joshua Taygerly[1], Kathleen Boyle[1], Stephen E. Basham[1], Chris Padovani[1,3], Yan Lou[1,4], Thomas J. Cummins[1], Stephanie L. Yung[1], Szerenke Kiss von Soly[1], Frank Kayser[1,5], John Kuriyan[2], Michael Rape[2], Mario Cardozo[1], Mark A. Gallop [1,6], Neil F. Bence[1], Paul A. Barsanti[1,6] & Anjanabha Saha [1,6]

Protein–protein interactions (PPIs) governing the recognition of substrates by E3 ubiquitin ligases are critical to cellular function. There is significant therapeutic potential in the development of small molecules that modulate these interactions; however, rational design of small molecule enhancers of PPIs remains elusive. Herein, we report the prospective identification and rational design of potent small molecules that enhance the interaction between an oncogenic transcription factor, β-Catenin, and its cognate E3 ligase, SCF$^{β-TrCP}$. These enhancers potentiate the ubiquitylation of mutant β-Catenin by β-TrCP in vitro and induce the degradation of an engineered mutant β-Catenin in a cellular system. Distinct from PROTACs, these drug-like small molecules insert into a naturally occurring PPI interface, with contacts optimized for both the substrate and ligase within the same small molecule entity. The prospective discovery of 'molecular glue' presented here provides a paradigm for the development of small molecule degraders targeting hard-to-drug proteins.

[1] Nurix Therapeutics, Inc., 1700 Owens Street, Suite 205, San Francisco, CA 94158, USA. [2] Department of Molecular and Cell Biology and Howard Hughes Medical Institute, University of California, Berkeley, CA 94720, USA. [3]Present address: Department of Molecular and Cell Biology, University of California, Berkeley, CA 94720, USA. [4]Present address: NiKang Therapeutics, Bldg. E500, 200 Powder Mill Road, Wilmington, DE 19803, USA. [5]Present address: BioArdis, 7310 Miramar Road San Diego, San Diego, CA 92126, USA. [6]Present address: 5AM Ventures, 501 2nd Street, Suite 350, San Francisco, CA 94107, USA. Correspondence and requests for materials should be addressed to K.R.S. (email: ksimonetta@nurix-inc.com) or to A.S. (email: sahaanjanabha@gmail.com)

Protein-protein interactions (PPIs) are central to most signaling pathways in cells and represent important nodes for therapeutic intervention. Although most small molecule PPI modulators are inhibitors, a small subset of PPI modulators are known to work by stabilizing PPIs[1,2]. These include immunosuppressants cyclosporine, FK506 and rapamycin as well as anti-cancer agents lenalidomide, pomalidomide, and thalidomide (collectively referred as immunomodulatory drugs or IMiDs). These IMiD molecules work by binding to the CRL4 E3 ligase substrate receptor protein, Cereblon (CRBN) and promoting interaction with the non-native lymphoid transcription factors Ikaros (IKZF1) and Aiolos (IFZF3), as well as Casein kinase 1α (CK1α) leading to their ubiquitylation by RING E3 ligase CRL4$^{CRBN}$ and subsequent degradation by the 26S proteasome[3–6]. This degradation contributes to the clinical efficacy of IMiDs. Although thalidomide was discovered over 70 years ago, the molecular mechanism of IMiDs as PPI enhancers were discovered only recently following careful characterization of their pharmacologically relevant targets[7,8]. In addition to the IMiDs, other ligase-binding molecules are known in plant physiology. In particular, the small molecule hormone auxin, also known as indole-3-acetic acid (IAA), regulates plant growth and development by modulating the recognition of Aux/IAA transcriptional repressors by the SCF$^{TIR1}$ E3 ligase[9]. Auxin binds to the substrate receptor protein TIR1 and promotes the interaction between TIR1 and the Aux/IAA substrate[10]. Importantly, auxin enhances the substrate interaction with TIR1 by extending the protein interaction interface for substrate binding, acting as a molecular glue by forming a continuous hydrophobic core for the TIR1 and substrate interaction[10].

The use of PPI enhancer molecules as 'molecular glue' to potentiate substrate:ligase interactions could be a therapeutically useful modality to employ against drug targets previously considered undruggable. However, rational design of such small drug-like molecules to degrade proteins that are prospectively identified remains elusive. In recent years, significant progress has been made to target protein degradation by employing hetero-bifunctional molecules (also referred as PROTACs or PROteolysis-TArgeting Chimeras)[11–14]. These bifunctional molecules consist of two small molecules connected by a linker. One small molecule binds a protein of interest (substrate) and the other binds an E3 ubiquitin ligase, thereby bringing the substrate in close proximity with the ligase, promoting its ubiquitylation and degradation. Although PROTACs represent important new chemotypes to induce substrate degradation, the molecular weight, physiochemical, and pharmaceutical properties fall outside of the typical range for small molecule drugs potentially leading to challenges in drug development. These challenges might be circumvented by discovering and rationally designing smaller molecular glue-like molecules that bind both the substrate (either native-substrate or neo-substrate) and the ligase without the need for a linker to induce substrate degradation.

Transcription factors remain extremely challenging proteins to target, despite being implicated in multiple diseases[15]. One such example is the Wnt signaling effector protein, β-catenin that is often dysregulated and stabilized in cancer[16,17]. In normal resting cells, the cytoplasmic pool of β-catenin is maintained at low levels through a process involving phosphorylation, ubiquitylation, and subsequent degradation. β-catenin is phosphorylated by the cytoplasmic destruction complex (DC) consisting of the tumor suppressor proteins, Axin and Adenomatous Polyposis Coli (APC), and two serine-threonine kinases, glycogen synthase kinase 3 (GSK3) and casein kinase 1 (CK1). β-catenin is initially phosphorylated at Ser45 by CK1, followed by GSK3 phosphorylation at Thr41, Ser37, and Ser33[18–21]. However, the sequential nature of the β-catenin phosphorylation cascade is not essential in certain colorectal cancer cells where phosphorylation at Ser45 is not required for phosphorylation at residues Ser33, Ser37, and Thr41[22]. The phosphorylated Ser33 and Ser37 in β-catenin are part of the phosphodegron sequence (DpSGφXpS) that binds to the F-box-containing E3 ubiquitin ligase protein β-TrCP, leading to β-catenin ubiquitylation by the SCF$^{β-TrCP}$ cullin-RING ligase and subsequent degradation by the 26S proteasome[23]. In the majority of colorectal cancers, β-catenin is stabilized either by inactivating mutations to APC or Axin, which prevent proper β-catenin phosphorylation or by mutations within the β-catenin phosphodegron sequence itself[24,25]. While disruption of β-catenin phosphorylation frequently accounts for elevated protein levels, mutations in β-TrCP are rarely observed[26]. These β-catenin phosphodegron mutations including Ser33 and Ser37, impair the ability of β-catenin to effectively bind to β-TrCP leading to its stabilization and thereby facilitating an enhanced oncogenic transcriptional program. Despite being an important biological target, no small molecule inhibitors of β-catenin are available as therapeutic agents[16,27]. Therefore, using the concept of molecular glue, we sought to prospectively discover small molecule β-catenin degraders that work by restoring mutant β-catenin binding to β-TrCP. This molecular glue strategy represents a unique approach to target tumors with elevated levels of β-catenin. More broadly, it represents a therapeutic strategy to target proteins defective in ligase binding, ubiquitylation and degradation that have been implicated in various diseases[28].

In this study, we report the prospective identification and rational structure-guided optimization of potent molecular glue-like small molecules intentionally designed to enhance the PPI between mutant β-catenin and β-TrCP that potentiate cellular degradation of an engineered mutant β-catenin in a cellular system. This work establishes a paradigm for rational design of small molecule enhancers to target ubiquitylation and degradation of proteins, including oncogenic transcription factors, by harnessing the ubiquitin proteasome system.

## Results

**Identification of β-catenin:β-TrCP interaction enhancers**. To characterize the interaction of mutant β-catenin with the ubiquitin ligase β-TrCP, we developed a fluorescence polarization (FP)-based binding assay utilizing β-catenin phosphodegron peptides (residues 17–48) and recombinant β-TrCP/Skp1 complex (referred as β-TrCP subsequently). Consistent with the role of the phosphodegron in targeting β-catenin to β-TrCP, the doubly-phosphorylated peptide (pSer33/pSer37) binds with 2 nM binding affinity whereas the non-phosphorylated peptide (Ser33/Ser37) binds with significantly lower affinity (>100 μM) (Fig. 1a). Removal of the single phosphate from Ser33 (Ser33/pSer37) results in a 10,000-fold loss of affinity (>10 μM) for β-TrCP. In contrast, removal of phosphate from Ser37 (pSer33/Ser37) results in a 300-fold loss in binding affinity relative to the doubly-phosphorylated peptide (Fig. 1a). Being able to quantitatively monitor the interaction of pSer33/Ser37 β-catenin to β-TrCP, we prioritized our molecular glue finding efforts on Ser37 mutants. The binding affinity of the pSer33/Ser37 peptide allowed development of a robust screening assay which could not be achieved for the much weaker affinity peptides lacking Ser33 phosphorylation. Ser37 is a known hotspot for β-catenin mutations (Supplementary Fig. 1a), representing ~10% of the known β-catenin mutations that are often mutated to Ala, Cys, and Phe[24]. These mutations at Ser37 impair the ability of monophosphorylated Ser33 β-catenin peptides to effectively bind β-TrCP by up to 300-fold compared to the doubly-phosphorylated β-catenin peptide (Supplementary Fig. 1b). Additionally, the ovarian endometrioid adenocarcinoma cell line, TOV-112D, which carries a

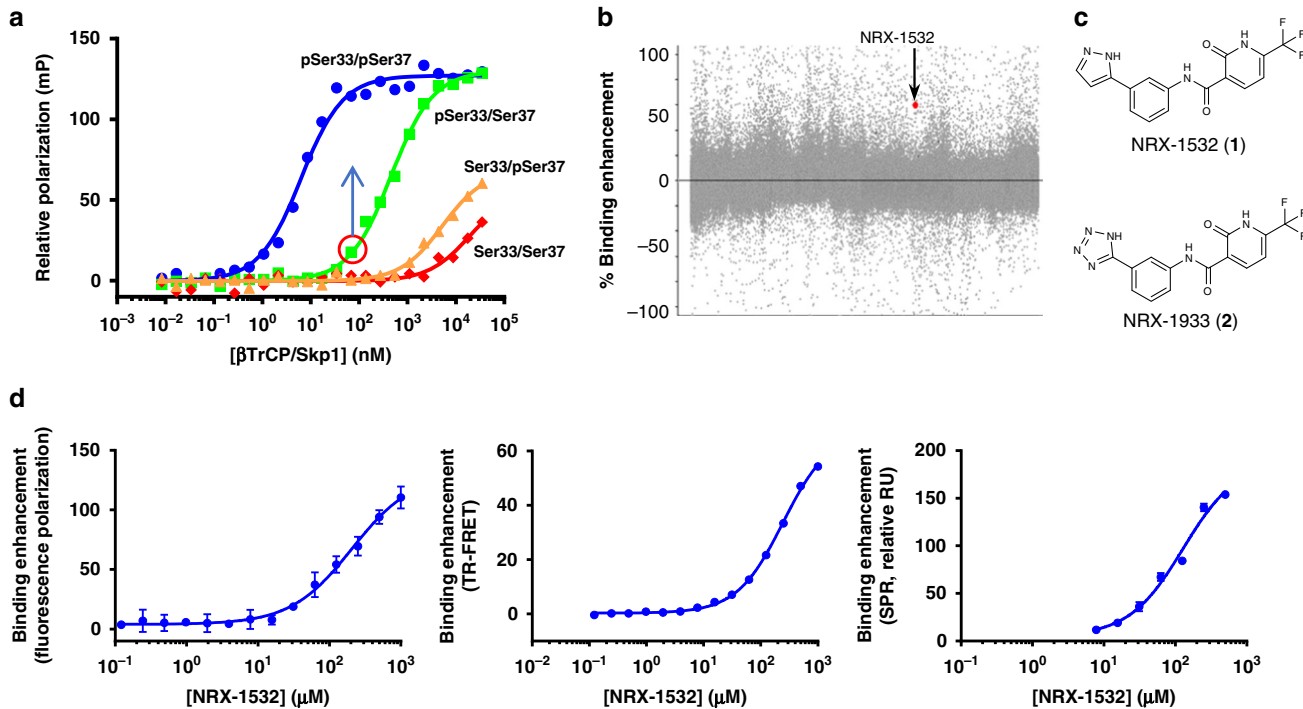

**Fig. 1** Identification of small molecule enhancers of β-catenin:β-TrCP interaction. **a** β-TrCP complex was titrated against 5 nM BODIPY-TMR-labeled β-catenin peptide (residues 17–48) of various phosphorylation status and binding was monitored by fluorescence polarization. Binding affinities were estimated by curve fitting to a one-site binding model. Circled on the pSer33/Ser37 β-catenin binding curve is the β-TrCP concentration that results in 20% binding saturation, which was used in the HTS. **b** Binding enhancement for all compounds screened in the HTS using the FP assay. Each dot represents an individual compound, with NRX-1532 highlighted. **c** Chemical structure of the enhancers, NRX-1532 and NRX-1933. **d** Enhancement of pSer33/Ser37 β-catenin peptide binding to β-TrCP with increasing concentrations of NRX-1532. In each assay, the concentration of either β-TrCP or β-catenin was adjusted to obtain 20% binding saturation in the absence of compound. Shown is a representative experiment, values reported are averages

homozygous S37A β-catenin mutation[29] is dependent on mutant β-catenin for viability (Supplementary Fig. 2a). Reduced binding of S37A mutant β-catenin to β-TrCP results in defective ubiquitylation in TOV-112D cells relative to HEK293T cells expressing WT β-catenin, leading to stabilization and extended half-life of endogenous S37A β-catenin in TOV-112D cells compared to WT β-catenin in HEK293T cells (Supplementary Fig. 2b, c).

Structural characterization of the β-catenin phosphodegron lacking Ser37 phosphorylation reveals that the monophosphorylated degron binds to β-TrCP in a similar conformation as the WT pSer33/pSer37 degron[30] (Supplementary Fig. 3a). Removal of the phosphate from Ser37 eliminates several key electrostatic and hydrogen bonding interactions between the β-catenin peptide and β-TrCP, leading to reduced binding affinity, but also reveals a small hydrophobic pocket between the β-catenin:β-TrCP interface (Supplementary Fig. 3b). We performed a high-throughput screen (HTS) on a 350,000 compound library utilizing the FP assay configured at 20% saturation of β-TrCP binding to the monophosphorylated pSer33/Ser37 β-catenin peptide. This HTS identified four structurally related compounds including NRX-1532 (1) that contains a 6-trifluoromethylpyridone core linked to a biaryl group via a secondary amide linkage (Fig. 1b, c). Further characterization of NRX-1532 confirmed that it enhances the pSer33/Ser37 β-catenin:β-TrCP interaction in a dose-dependent manner in the FP assay with a potency (EC$_{50}$) of 206 ± 54 μM, as well as in orthogonal TR-FRET and SPR binding assays with similar EC$_{50}$ values of 246 ± 17 μM and 129 ± 33 μM, respectively (Fig. 1d).

To elucidate the binding mode of the enhancer molecule, we solved the crystal structure of the ternary complex of a related but more soluble tetrazole analog, NRX-1933 (2), bound to the β-catenin:β-TrCP complex (Fig. 2a). The structure reveals that NRX-1933 binds at the β-catenin:β-TrCP interface with the trifluoromethylpyridone occupying the small pocket that is revealed by the absence of Ser37 phosphorylation (Fig. 2b). Enhancer binding to the complex has minimal impact on the conformation of either β-catenin or β-TrCP. The trifluoromethyl substituent fills the small hydrophobic pocket formed by the Leu31 and Ile35 residues of the β-catenin peptide and by the Ala434 and Leu472 residues of β-TrCP, forming important hydrophobic interactions with both β-catenin and β-TrCP (Fig. 2c). The trifluoromethylpyridone is anionic (pK$_a$ of NRX-1532 is 5.6) and functions as a phosphate mimetic that occupies the β-catenin Ser37 phosphate binding site and mimics the phosphate by forming a hydrogen bond with the backbone N–H of Gly432 on β-TrCP. Lastly, the phenyltetrazole tail group extends away from the β-catenin:β-TrCP interface, with the phenyl ring stacking against Arg431 of β-TrCP and the anionic tetrazole residing in close proximity to the cationic Arg410 and Arg431 residues of β-TrCP (Fig. 2d and Supplementary Fig. 3c).

To determine the degree to which NRX-1532 enhances the binding affinity of β-TrCP for the monophosphorylated degron peptide (referred to as the cooperativity of compound binding), we monitored the binding affinity of β-catenin to the β-TrCP at increasing concentrations of compound (Fig. 2e). In a TR-FRET-based binding assay, NRX-1532 enhances the binding of pSer33/Ser37 β-catenin to β-TrCP with 10-fold cooperativity, increasing the binding affinity from 689 nM in the absence of compound to 68 nM at 500 μM compound. The increase in binding affinity plateaus at high concentrations, indicating saturation of the NRX-1532:β-catenin:β-TrCP ternary complex formation. Importantly,

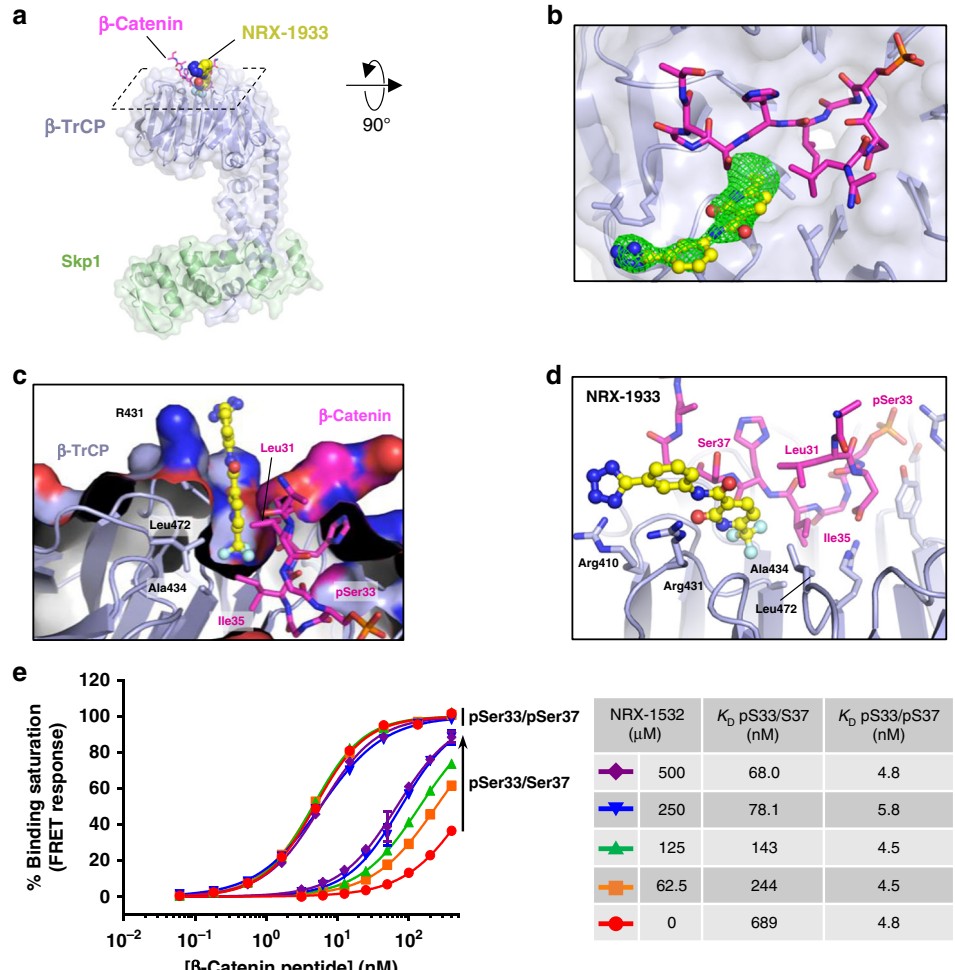

**Fig. 2** NRX-1933 binds at the pSer33/Ser37 β-catenin:β-TrCP interface and potentiates the binding affinity of the interaction. **a** Overall structure of NRX-1933 (yellow spheres) bound to the β-catenin:β-TrCP complex (PDB: 6M93) (Supplementary Table 1). **b** Unbiased Fo-Fc electron density for NRX-1933 at 3.0 standard deviations above the mean is shown (green). The compound is shown as sticks (yellow), bound between the surface of the β-TrCP β-propeller (gray) and the monophosphorylated β-catenin degron peptide, shown as sticks (magenta). **c** Cutaway view of the hydrophobic pocket formed by residues Leu472 and Ala434 from β-TrCP and Ile35 and Leu31 from β-catenin. **d** Side view of the same interaction with the β-TrCP surface removed, highlighting the β-catenin and β-TrCP residues that interact with NRX-1933. **e** The binding affinities of the pSer33/Ser37 and pSer33/pSer37 β-catenin peptides for β-TrCP were measured in the TR-FRET assay by titrating BODIPY-FL-labeled β-catenin peptides (residues 17–48) against 300 pM β-TrCP at varying concentrations of NRX-1532. The binding curves for the pSer33/pSer37 β-catenin peptide titrations were fitted to one-site binding model. For the pSer33/Ser37 β-catenin peptide titrations, which do not reach saturation owing to limitations in the maximum amount of fluorophore compatible with this assay. Curves were fitted to the same one-site binding model, with the upper plateau constrained to that of the pSer33/pSer37 β-catenin peptide binding curve. The $K_D$s are reported for both pSer33/Ser37 and pSer33/pSer37 β-catenin peptides

the binding affinity of doubly-phosphorylated pSer33/pSer37 β-catenin to the β-TrCP was unaffected by the compound, implying that these enhancers are specific for β-catenin defective in Ser37 phosphorylation. Similar enhancement of binding affinity was observed in an orthogonal FP binding assay (Supplementary Fig. 4a). Additionally, these enhancer molecules are specific for β-catenin and do not potentiate the binding of other β-TrCP substrates, such as Emi1, IκBα, and Wee1 (Supplementary Fig. 4b). To evaluate the ability of these enhancer molecules to potentiate ubiquitylation of pSer33/Ser37 β-catenin by the β-TrCP complex, we established a functional ubiquitylation assay utilizing a β-catenin peptide spanning residues 17–60 and recombinant SCF$^{β-TrCP}$. Consistent with the defect in β-TrCP binding, the pSer33/Ser37 β-catenin peptide displayed reduced ubiquitylation relative to the doubly-phosphorylated peptide. Importantly, NRX-1532 weakly enhanced the ability of β-TrCP to ubiquitylate the pSer33/Ser37 β-catenin peptide (Supplementary Fig. 4c).

**Optimized enhancers have improved potency and cooperativity**. The small molecule enhancers NRX-1532 and NRX-1933 have reasonable molecular weight and drug-like physiochemical properties to serve as starting points for a structure-enabled design campaign to improve both potency and cooperativity (Supplementary Table 2). Optimizing this system is challenging because of the small volume of the enclosed binding site[31], the cationic nature of the phosphate binding site, and the increased entropic penalty of forming a ternary complex. However, analysis of the bound NRX-1933 crystal structure b-factors indicates that the N-terminus of the β-catenin peptide has the potential for movement. Therefore, we initially focused on strategies to induce movement in this region of β-catenin to induce a larger, more druggable binding pocket. This strategy was realized early in the design campaign, with analogues such as NRX-2663 (3), where phenoxy substitution at the 4-position of the trifluoromethyl pyridone core induces rearrangement of β-catenin Leu31 to create a larger binding pocket (Fig. 3a, b and Supplementary Fig. 5a).

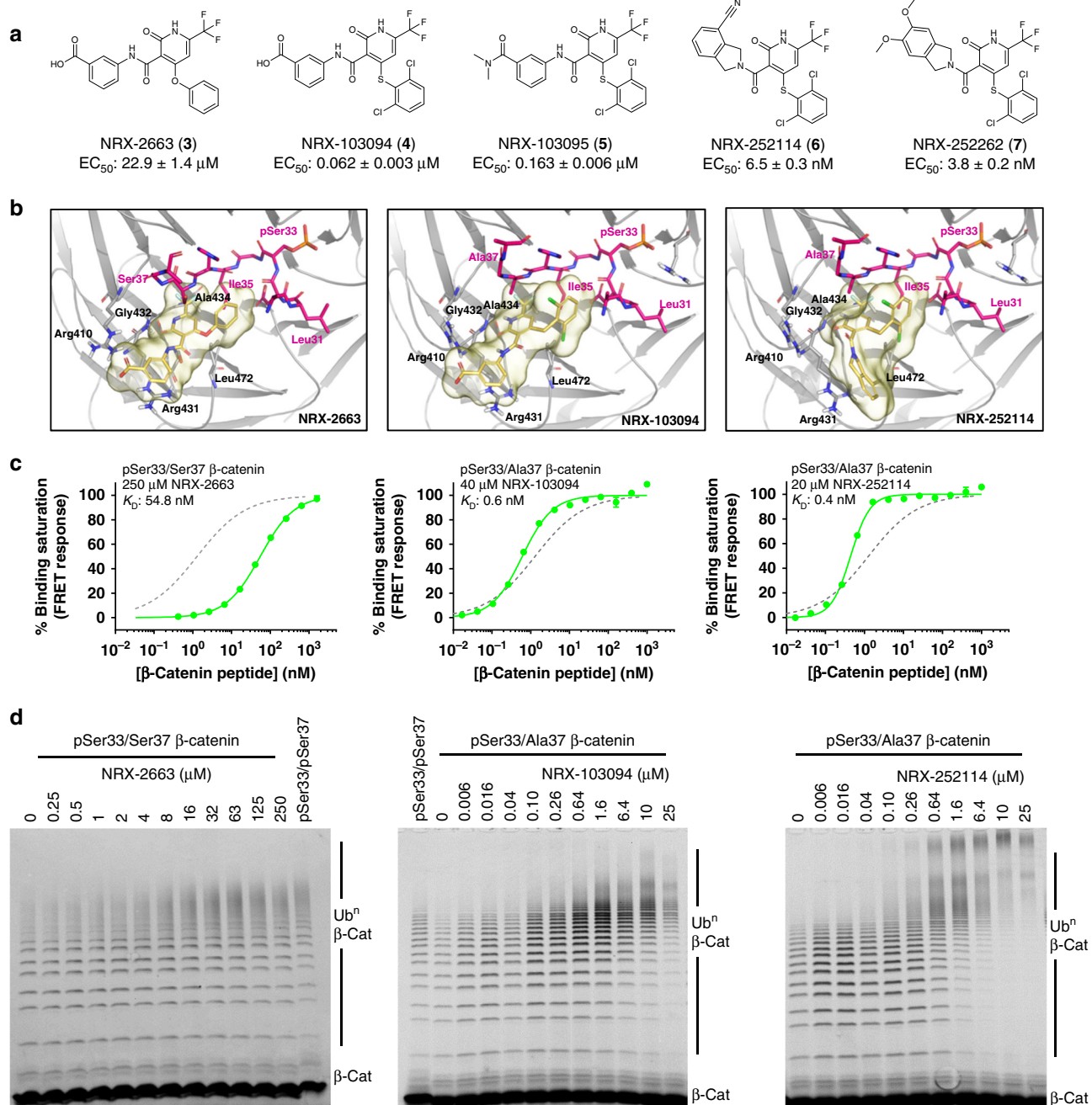

**Fig. 3** Optimized enhancers potentiate pSer33/S37A β-catenin binding and ubiquitylation. **a** Chemical structure of optimized enhancers and their potency in TR-FRET assay against pSer33/S37A β-catenin peptide for β-TrCP binding. **b** Structures of enhancers (yellow) in complex with β-catenin peptide (magenta) and β-TrCP (gray). The binding poses and β-catenin degron conformations shown for NRX-2663 (PDB: 6M92) and NRX-103094 (6M91) are from crystal structures (Supplementary Table 1), whereas the NRX-252114 pose was determined by computational docking. **c** Binding affinities of the β-catenin peptides (as indicated) for β-TrCP were measured in the TR-FRET assay in presence of enhancers at concentration indicated. For NRX-252114, the binding assay is bottoming out due to 300 pM β-TrCP concentration, with the curve fit yielding a higher than expected hill slope of 1.8. Therefore, the $K_D$ reported for NRX-252114 represents an upper estimate. For reference, the binding curve for pSer33/pSer37 β-catenin peptide in the same TR-FRET assay is represented in dotted gray line with a $K_D$ of 2 nM. **d** 4 μM of fluorescently-labeled β-catenin peptide (residues 17–60; as indicated) was ubiquitylated in the presence of 125 nM Ube1, ATP, 1.75 μM Cdc34 and 100 nM SCF$^{β\text{-TrCP}}$ with increasing concentration of enhancers. β-Catenin ubiquitylation were resolved by SDS-PAGE and imaged for fluorescence

NRX-2663 displays both improved potency in the binding assay with pSer33/Ser37 β-catenin (EC$_{50}$ = 80 ± 4 μM) and cooperativity of binding (13-fold, Fig. 3c). NRX-2663 also markedly enhances the ubiquitylation of the pSer33/Ser37 β-catenin peptide (Fig. 3d), with ubiquitin chain formation at ligand concentrations of 16 μM and above resembling ubiquitin chain formation on WT

pSer33/pSer37 β-catenin peptide. Further optimization of the phenoxy substituent gave diarylthioether derivatives NRX-103094 (4) and NRX-103095 (5) with significant improvements in potency (EC$_{50}$ = 457 ± 23 nM for NRX-103094) with pSer33/Ser37 β-catenin peptide (Fig. 3b and Supplementary Fig. 5b). Additionally, these molecules broadly enhanced the peptide

binding of all the prevalent Ser37 mutations, including phenylalanine, cysteine, and alanine (Supplementary Table 2). Since these enhancers show the best activity against the β-catenin S37A mutant, and because of the ready access to the homozygous S37A mutant line TOV-112D, we prioritized our efforts on this isoform. With pSer33/S37A β-catenin, NRX-103094 gives an $EC_{50}$ of $62 \pm 3$ nM and displays a 1000-fold cooperativity of peptide binding to β-TrCP (Fig. 3c). Importantly, NRX-103094 potentiates the ubiquitylation of pSer33/S37A β-catenin peptide to levels greater than WT pSer33/pSer37 levels at 250 nM (Fig. 3d). Introduction of the cyanoisoindoline in NRX-252114 (6) resulted in another significant gain in potency ($EC_{50} = 6.5 \pm 0.3$ nM). Computational docking of NRX-252114 bound to the β-catenin: β-TrCP complex suggested that the cyanoisoindoline projects out of the plane from the trifluoromethylpyridone and makes additional hydrophobic contacts with β-TrCP. NRX-252114 enhances the binding of pSer33/S37A β-catenin peptide for β-TrCP with >1500-fold cooperativity, surpassing the affinity of WT doubly-phosphorylated peptide (Fig. 3c). However, the full extent of cooperativity could not be measured in this assay since the $K_D$ of the peptide in the presence of compound is lower than the concentration of β-TrCP in the assay, resulting in stoichiometric binding and a curve hill slope > 1. Thus, the $K_D$ represents only a lower limit of the binding affinity. Corresponding to improvement in binding affinity, NRX-252114 further enhances the ubiquitylation of pSer33/S37A β-catenin peptide to form long ubiquitin chains (Fig. 3d). Finally, another isoindoline analog, NRX-252262 (7) with dimethoxy substitution displays slightly improved potency ($EC_{50} = 3.8 \pm 0.2$ nM). Compared to the original HTS hit NRX-1532, with an $EC_{50} > 200$ μM and only eight-fold cooperativity, NRX-252114 and NRX-252262 represent significant improvements in both measures. Importantly, improvements in both potency and cooperativity translated into enhanced ubiquitylation of mutant β-catenin peptide, with enhancer molecules potentiating ubiquitin chain formation to levels greater than WT β-catenin peptide.

**Enhancers potentiate unphosphorylated β-catenin ubiquitylation.** The optimized enhancers NRX-252114 and NRX-252262 had suitable drug-like properties for cellular evaluation (Supplementary Table 2); however, they did not induce degradation of endogenous S37A mutant β-catenin in TOV-112D cells. We reasoned that the lack of degradation could be due to insufficient Ser33 phosphorylation in S37A mutant β-catenin (Supplementary Fig. 6a), which is consistent with the sequential nature of β-catenin phosphorylation[18,21]. Since the non-phosphorylated Ser33/S37A β-catenin could potentially be the prevalent β-catenin species in the TOV-112D cells, we evaluated the magnitude of the defect in non-phosphorylated Ser33/S37A β-catenin binding to β-TrCP and determined how effectively these enhancers potentiate this interaction. The non-phosphorylated Ser33/Ser37 peptide binds β-TrCP with very weak affinity (>100 μM); however, in the presence of 20 μM NRX-252114 that binding affinity is enhanced to 180 nM (Fig. 4a). This represents a >500-fold enhancement in PPI binding affinity, although this is still 90-fold lower than WT pSer33/pSer37 β-catenin peptide binding to β-TrCP. Consistent with binding enhancement, NRX-252114 also potentiates the ubiquitylation of unphosphorylated Ser33/S37A β-catenin peptides by the SCF$^{β\text{-}TrCP}$ complex (Fig. 4b). Interestingly, in the presence of NRX-252114, unphosphorylated β-catenin peptides are ubiquitylated with chain lengths similar to WT doubly-phosphorylated peptide; however, only a small percentage of unphosphorylated β-catenin is ubiquitylated. Consistent with peptide ubiquitylation, these enhancers also potentiate the ubiquitylation activity of full-length mutant Ser33/

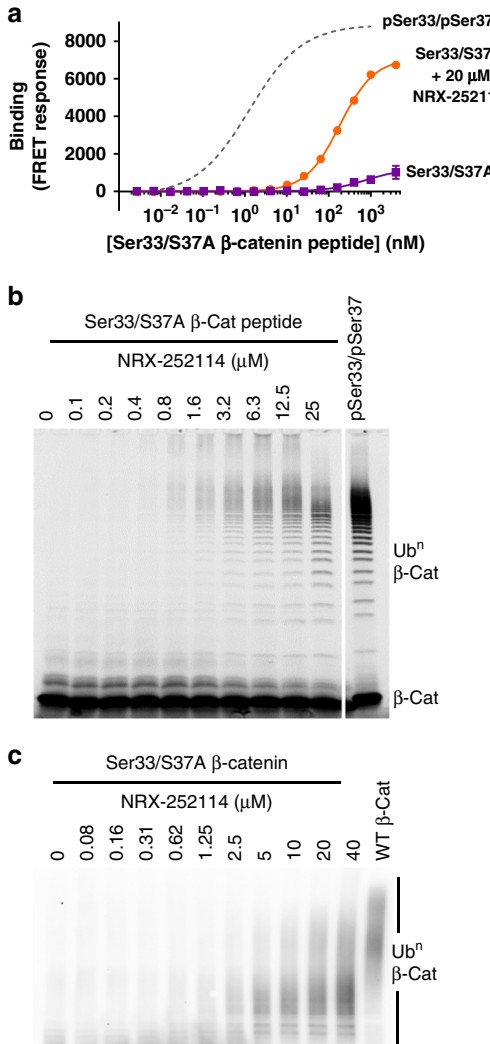

**Fig. 4** Enhancers potentiate binding and ubiquitylation of the unphosphorylated β-catenin degron. **a** Binding of Ser33/S37A β-catenin peptide for β-TrCP measured in the TR-FRET assay in absence or presence of 20 μM NRX-252114. For reference, the binding curve for pSer33/pSer37 β-catenin peptide in the same TR-FRET assay is represented in dotted gray line. **b** Ubiquitylation of Ser33/S37A β-catenin peptide (residues 17–60) by SCF$^{β\text{-}TrCP}$ in the presence of NRX-252114 under conditions similar to Fig. 3d. **c** 400 nM purified full-length S37A mutant β-catenin protein was ubiquitylated in the presence of 125 nM Ube1, ATP, 1.75 μM Cdc34, and 400 nM SCF$^{β\text{-}TrCP}$ with the increasing concentration of NRX-252114. Reactions were resolved by SDS-PAGE and analyzed by immuno-blotting with a β-catenin C-terminal antibody. For reference, ubiquitylation of WT β-catenin protein is also shown (* refers to degradation products)

S37A β-catenin protein with the more potent NRX-252114 displaying a higher degree of ubiquitylation than NRX-103094 (Fig. 4c and Supplementary Fig. 6b, c). In the presence of 40 μM NRX-252114, the level of ubiquitylation activity on Ser33/S37A mutant β-catenin approaches that of WT full-length protein (Fig. 4c). The robust binding and ubiquitylation enhancement of unphosphorylated Ser33/S37A β-catenin with NRX-252114 in biochemical assays suggests that unphosphorylated β-catenin might be degraded in cellular system with further improvement of these enhancer molecules.

**Induced cellular degradation of phosphomimetic β-catenin.** The lack of cellular activity in TOV-112D cells suggested that non-phosphorylated Ser33/S37A might be the prevalent β-catenin species in the TOV-112D cells. To overcome this lack of Ser33 phosphorylation, we developed a S33E phosphomimetic system to evaluate the cellular activity of these enhancers. In this system, the Ser33 status of β-catenin is not impacted by the sequential nature of the phosphorylation cascade (Fig. 5a). Replacement of Ser33 with glutamic acid (S33E/S37A β-catenin) results in a >10-fold decrease in the binding affinity for β-TrCP relative to the

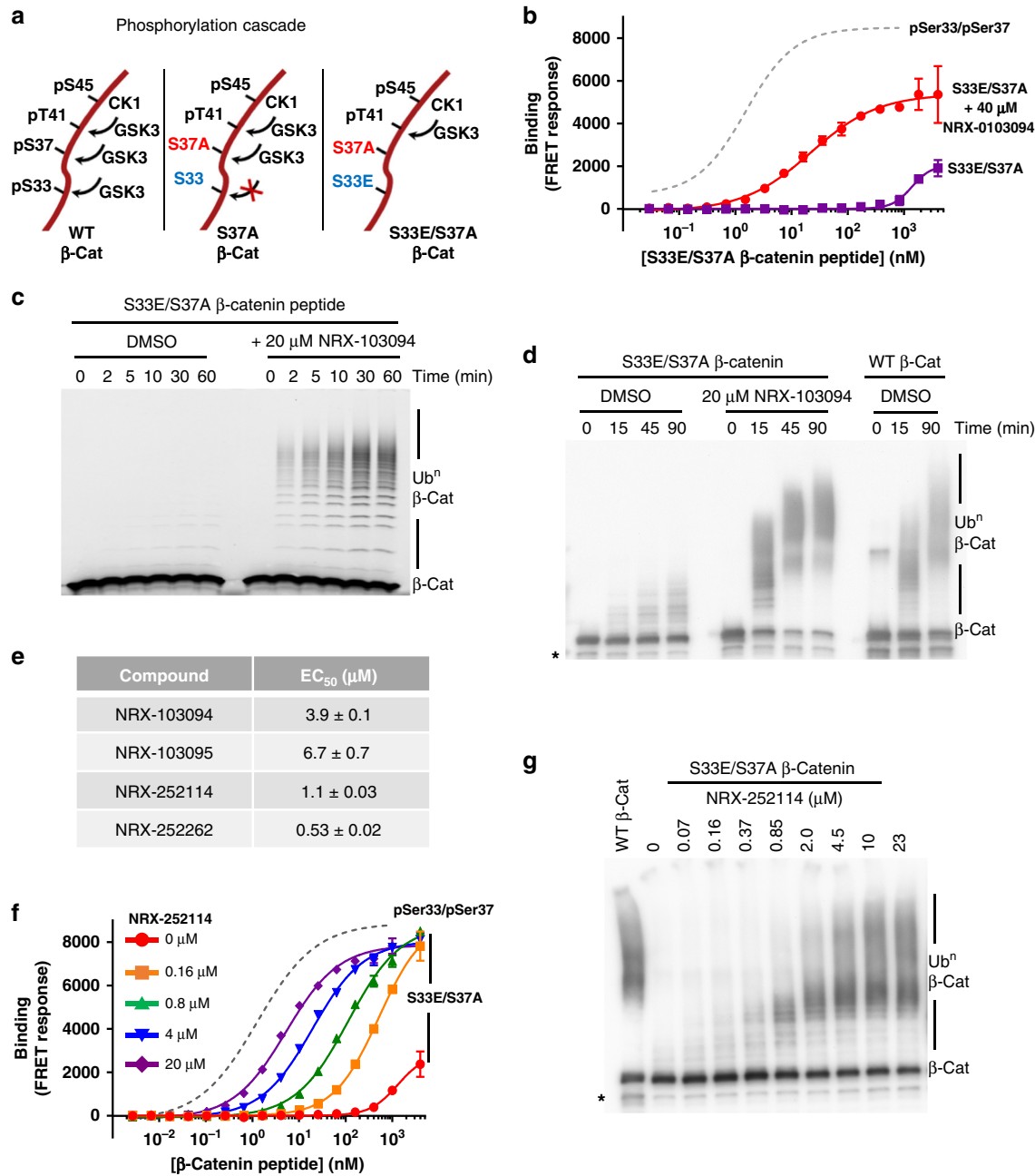

**Fig. 5** Enhancers potentiate binding and ubiquitylation of S33E/S37A phosphomimetic β-catenin. **a** Schematic of the proposed phosphorylation cascade for WT and S37A, S33E/S37A mutant β-catenin. **b** Binding curve of S33E/S37A β-catenin peptide for β-TrCP measured in the TR-FRET assay in absence or presence of 40 μM NRX-103094. For reference, the binding curve for pSer33/pSer37 β-catenin peptide in the same TR-FRET assay is represented in dotted gray line. **c** 4 μM of fluorescently-labeled S33E/S37A β-catenin peptide (residues 17–60) was ubiquitylated in the presence of 125 nM Ube1, ATP, 1.75 μM Cdc34, 100 nM SCF$^{β-TrCP}$, and 20 μM NRX-103094. Reactions were resolved by SDS-PAGE and imaged for fluorescence. **d** 400 nM purified full-length β-catenin protein (as indicated) was ubiquitylated in the presence of 125 nM Ube1, ATP, 1.75 μM Cdc34, 400 nM SCF$^{β-TrCP}$, and 20 μM NRX-103094. Reactions were resolved by SDS-PAGE and analyzed by immuno-blotting with a β-catenin C-terminal antibody (* refers to degradation products). **e** Enhancer potencies for S33E/S37A β-catenin peptide binding to β-TrCP. Due to low intrinsic binding affinity, the pSer33/pSer37 β-catenin binding assays were carried out at 1% binding saturation. **f** Binding curve of S33E/S37A β-catenin peptide for β-TrCP measured in the TR-FRET assay in presence of varying NRX-252114 concentration. For reference, the binding curve for pSer33/pSer37 β-catenin peptide in the same TR-FRET assay is represented in dotted gray line. **g** Ubiquitylation of full-length S33E/S37A β-catenin protein in the presence of varying concentration of NRX-252114 (as described above; * refers to degradation products)

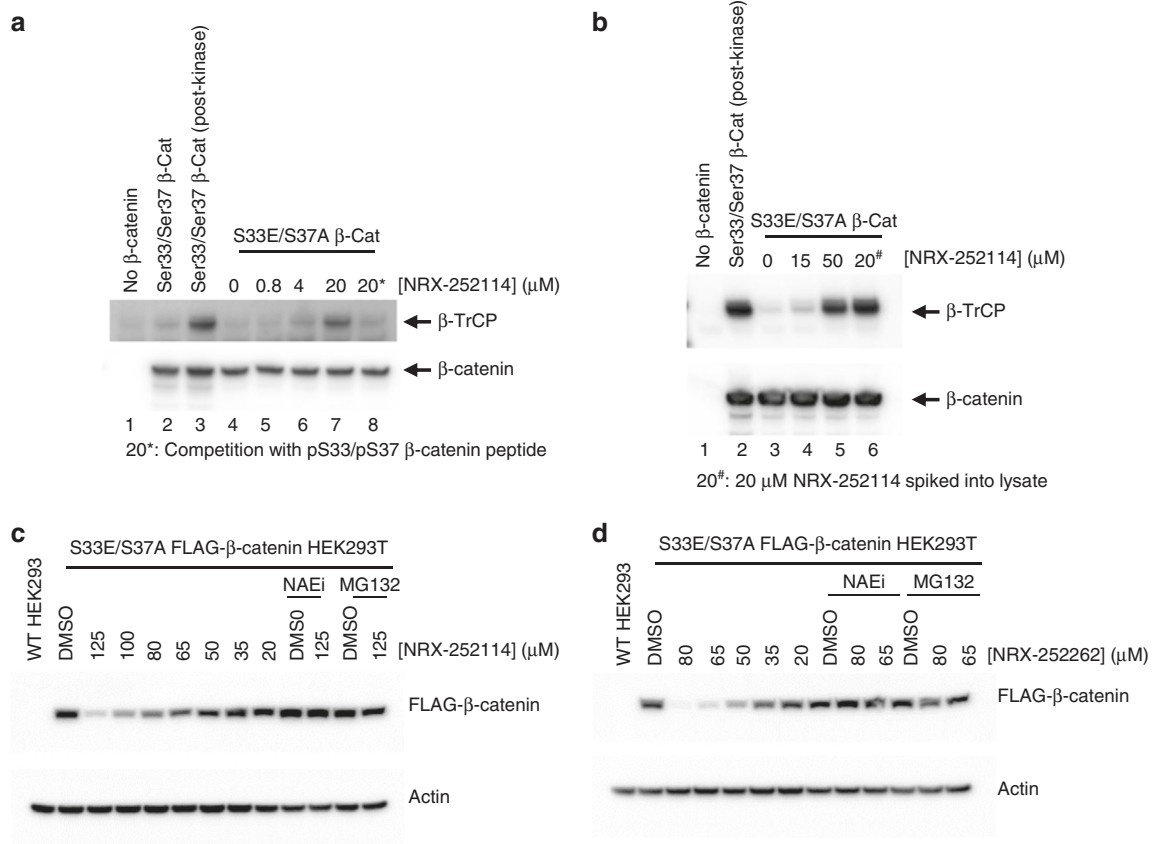

**Fig. 6** Enhancers degrade S33E/S37A phosphomimetic β-catenin protein in cells. **a** Purified Myc/FLAG-tagged β-catenin protein (as indicated) was mixed with HEK293T cell lysates in the presence of varying amounts of NRX-252114 and immunoprecipitated using anti-myc beads. The samples were resolved on SDS-PAGE and analyzed by immuno-blotting with either C-terminal β-catenin or β-TrCP antibody. In lane 3, 2 μM β-catenin was phosphorylated with the mix of 1 μM GSK3, 200 nM CK1 and 50 nM Axin prior to its addition to the lysate. In lane 8, 6.4 μM pSer33/pSer37 β-catenin peptide was spiked into the lysate prior to the addition of β-catenin protein. **b** HEK293T cells without (lanes 1 and 2) or with overexpression of S33E/S37A Myc/FLAG-tagged β-catenin (lanes 3-6) were treated with NRX-252114 for 6 h (as indicated). Cells were then lysed, β-catenin was immunoprecipitated using anti-myc beads and probed for either β-catenin or β-TrCP (as described above). In lane 2, cell lysate was spiked with phosphorylated WT Myc/FLAG-tagged β-catenin protein, whereas in lane 6, the lysate was spiked with 20 μM NRX-252114 as positive controls. **c** HEK293T cells stably expressing Myc/FLAG-tagged S33E/S37A β-catenin were treated with varying concentrations of NRX-252114 (as indicated) for 6hrs and examined for S33E/S37A β-catenin levels by western analysis using anti-FLAG antibody. For control, Actin levels were monitored using anti-Actin antibody. For co-treatments, either 5 μM of Nedd8 E1 inhibitor (NAEi) or 10 μM of proteasome inhibitor (MG132) were added to cells 30 min prior to the compound treatment. **d** Same as Fig. 6c, except NRX-252262 was used

pSer33/S37A peptide (we could not estimate the binding affinities due to partial binding curves). In this system, NRX-103094 enhances the S33E/S37A phosphomimetic peptide binding affinity for β-TrCP from >5 μM in the absence of compound to 22 nM in the presence of 40 μM compound (Fig. 5b), which is similar to the binding enhancement observed for pSer33/S37A peptide. Consistent with the binding affinities, S33E/S37A β-catenin peptide had limited ubiquitylation by the SCF$^{β-TrCP}$ complex in the absence of compound. However, in the presence of 20 μM NRX-103094, robust time-dependent ubiquitylation of S33E/S37A β-catenin peptide is observed (Fig. 5c). Likewise, the full-length S33E/S37A β-catenin protein is strongly ubiquitylated in the presence of 20 μM NRX-103094, with the ubiquitylation levels approaching that of WT full-length β-catenin protein (Fig. 5d). The potency trends observed in the enhancer compounds for pSer33/S37A β-catenin peptide binding to β-TrCP translated proportionately for the S33E/S37A phosphomimetic peptide binding. However, the potencies were consistently diminished with the S33E/S37A peptide (Fig. 5e). Twenty-micromolar NRX-252114 enhanced the binding affinity of the S33E/S37A β-catenin peptide for β-TrCP to 6 nM, which is within

three-fold of that of WT doubly-phosphorylated peptide (Fig. 5f). Importantly, NRX-252114 also markedly enhances the ubiquitylation of S33E/S37A β-catenin protein (Fig. 5g), with the ubiquitin chain formation at ligand concentrations of 10 μM and above resembling that of WT β-catenin protein.

We then examined whether these enhancers can potentiate the formation of the mutant β-catenin:β-TrCP interaction in cell lysates. Purified myc-tagged β-catenin protein was combined with HEK293T cell lysates, immunoprecipitated using anti-myc beads, and probed for their ability to pull down endogenous β-TrCP. As expected, WT β-catenin pulled down β-TrCP only upon phosphorylation by GSK3, CK1, and Axin (Fig. 6a, lanes 2 and 3). No S33E/S37A mutant β-catenin:β-TrCP interaction was observed in absence of enhancer (Fig. 6a, lane 4). In contrast, addition of NRX-252114 to the cell lysate resulted in a dose-dependent interaction of mutant β-catenin to β-TrCP (Fig. 6a, lanes 5–7). Importantly, this interaction was specific since the addition of pSer33/pSer37 β-catenin peptide was able to effectively compete the formation of the NRX-252114:β-catenin: β-TrCP ternary complex (Fig. 6a, lane 8). To further establish the formation of the ternary complex in cells, myc-tagged β-catenin

constructs were transiently transfected in HEK293T cells and evaluated for their ability to pull down endogenous β-TrCP. Consistent with the findings in cell lysates, 50 μM NRX-252114 treatment resulted in robust S33E/S37A β-catenin:β-TrCP interaction in the transfected HEK293T cells (Fig. 6b, lanes 3-5).

These experiments confirmed that NRX-252114 is competent to form a ternary complex with the full-length S33E/S37A β-catenin and β-TrCP in cells. We then examined whether these enhancers can potentiate the degradation of S33E/S37A mutant β-catenin in cells. To test this, we established an engineered HEK293T cell line stably expressing the S33E/S37A phosphomimetic mutant β-catenin. As expected, the mutant S33E/S37A β-catenin was stable relative to WT β-catenin (Supplementary Fig. 7a). In agreement with the biochemical results, NRX-252114 treatment of phosphomimetic expressing cells resulted in a dose-dependent degradation of S33E/S37A mutant β-catenin starting at ~50 μM (Fig. 6c). Importantly, the degradation of S33E/S37A mutant β-catenin can be completely blocked by either inactivation of the ubiquitylation pathway using the Nedd8 E1 inhibitor, MLN4924, or by inactivation of the degradation pathway using the proteasome inhibitor, MG132 (Fig. 6c). Furthermore, this cellular degradation of β-catenin is dependent on β-TrCP, as siRNA knockdown of β-TrCP eliminates the degradation of S33E/S37A mutant β-catenin (Supplementary Fig. 7b). The protein levels of other β-TrCP substrates including WT β-catenin and IκBα remained unchanged. The more potent NRX-252262 also induces S33E/S37A mutant β-catenin degradation at an even lower concentration of ~35 μM (Fig. 6d). Collectively, these results demonstrate that these enhancers act as molecular glue in cells to promote the ubiquitylation and subsequent degradation of mutant β-catenin by the 26S proteasome. This provides an enticing path towards targeting β-catenin over-expressing cancer cells.

## Discussion

Here we report the prospective discovery and rational design of small molecule molecular glues that enhance a substrate:ligase interaction to induce substrate degradation. These molecules potentiate the binding of S37A mutant β-catenin to β-TrCP. The enhanced PPI affinity results in increased K48-linked ubiquitylation of mutant β-catenin by its natural ubiquitin ligase SCF[β-TrCP], thereby promoting its proteasomal degradation. Since S37A mutant β-catenin is defective in β-TrCP binding, improving the cooperativity was essential to overcome the binding defect and restore functional ubiquitylation in the mutant system. Strategically, we prioritized molecular modifications that were contained at the β-catenin:β-TrCP interface rather than growing the enhancer molecule away from the protein–protein interface. This strategy allowed concomitant improvements in both the cooperativity (>1500-fold) and potency (>10,000-fold) of the enhancers. This required improving affinity by increasing ligand efficiency, rather than improving affinity by increasing ligand molecular weight and growing away from the PPI interface. This allowed the enhancers to retain drug-like physiochemical properties throughout the optimization process even though they evolved from a weak screening hit.

While a few retrospectively characterized enhancers of substrate:ligase interactions have been reported[3,4,6–8,32–34] this work represents a prospective discovery of such molecules for a specific ligase-substrate pair. Compared to the better characterized enhancers, auxin and the IMiDs that directly bind to their respective ligases to create surfaces for substrate binding[7,8,10], these enhancers do not exhibit appreciable binding directly to β-TrCP alone and only bind in the context of the ternary complex. The enhancers herein also differ from the other known substrate:

ligase enhancers by promoting the interaction of β-TrCP with its native substrate, β-catenin, that has otherwise lost affinity because of its mutational status. The enhancer molecules provide the essential binding surface between β-catenin and β-TrCP that is lost as a result of the S37A β-catenin mutation. In contrast, the IMiD enhancers promote the binding of neo-substrates IKZF1, IKZF3, CK1α, and GSPT1 to the ubiquitin ligase CRBN[3–8,32], while another enhancer, indisulam, promotes the binding of neo-substrate Rbm39 to the ubiquitin ligase DCAF15[33,34]. These previously reported substrates have no endogenous connection to the ligase and display no binding affinity in the absence of enhancers. Compared to hetero-bifunctional molecules, these enhancers have an acceptable molecular weight for small molecules with desirable drug-like properties. Rather than degrading substrates with bifunctional molecules by hijacking an unrelated ligase where tissue-specific expression of both proteins could be a challenge to drive ternary complex formation for efficient ubiquitylation, these enhancers co-opt the native ligase to degrade substrate proteins that has otherwise lost binding due to mutations. Additionally, utilizing a native substrate-ligase pair that enhances PPI through the naturally occurring interface ensures both cooperative interaction as well as provide accessible substrate lysines for efficient ubiquitylation.

Developing small molecule therapeutics targeting the Wnt/β-catenin signaling pathway remains a challenge despite extensive efforts to target various components of the pathway[27]. Our work provides an approach to target the Wnt signaling pathway by promoting degradation of mutant β-catenin by SCF[β-TrCP]. Since these enhancer ligands bind at the phosphoserine binding site on β-TrCP, binding of the doubly-phosphorylated WT β-catenin is not enhanced by these ligands. This ensures the specificity of the enhancers for the mutant form of β-catenin without impacting the function of WT β-catenin in normal tissues. Due to the uncertainty of β-catenin Ser33 phosphorylation status in the mutant TOV-112D cells, we utilized an engineered Ser33 phosphomimetic β-catenin model to demonstrate cellular mechanistic proof-of-concept for mutant β-catenin degradation. Additionally, the small molecule NRX-252114 enhanced the affinity of unphosphorylated Ser33/S37A β-catenin for β-TrCP from >10,000-fold to within 90-fold of WT doubly-phosphorylated β-catenin affinity. Importantly, these enhancers potentiated unphosphorylated β-catenin ubiquitylation with chain lengths similar to that of doubly-phosphorylated WT β-catenin. This tantalizing result suggests that further potency improvements to the small molecule could drive unphosphorylated β-catenin ternary complex formation in cellular and in vivo contexts, and enable degradation of unphosphorylated β-catenin in β-catenin dependent tumors typically observed in cancers with APC, Axin or β-catenin mutations. This work shows that a prospective discovery approach can be rationally applied to develop small molecule β-catenin degraders beyond Ser37 mutations.

Transcription factors such as β-catenin are implicated in the development of many diseases, but with few exceptions, effectively remain as an undruggable target class. It is now evident that post-translational modification via the ubiquitin proteasome system is a common mechanism by which the levels of transcription factors are regulated in cells[15,28]. Dysregulation often results from mutations leading to compromised binding of transcription factors to their cognate ligases. This work provides a rational framework to prospectively employ small molecules that target such oncogenic transcription factors by enhancing substrate:ligase interactions to promote their degradation. Ultimately, this strategy could be useful to target therapeutically important classes of proteins that were previously deemed undruggable and might find further application to other PPIs of different classes as well.

## Methods

**Constructs and protein purification.** β-TrCP/Skp1 constructs[30] were cloned and co-expressed in *E. coli* in pCDF-Duet1 vector as 6xHis-SMT3-β-TrCP1/Skp1 (for FP) or as 6xHis-SMT3-β-TrCP1-avi tag/Skp1 (for TR-FRET; co-expressed with BirA, Avidity). Cultures were grown at 37 °C to an $OD_{600}$ of 1.0 and then induced for 16 h with 1 mM IPTG at 18 °C. 0.5 mM d-biotin was also added to cultures containing avi-tagged β-TrCP and BirA at the time of induction. Bacteria were pelleted and lysed in buffer containing 50 mM $NaH_2PO_4$ pH 8.0, 500 mM NaCl, 20 mM Imidazole, 5 mM BME, and protease inhibitors (0.2 mM AEBSF, 0.005 mM Leupeptin, and 0.5 mM Benzamidine). The lysate was incubated with Ni-NTA agarose (Qiagen), washed, and eluted with buffer containing 250 mM imidazole. Eluate was cleaved by ULP1, buffer exchanged overnight, and re-purified by Ni-NTA agarose to remove the cleaved 6xHis-SMT3 as well as any un-cleaved protein. The unbound fraction was further buffer exchanged into low salt, low pH buffer (50 mM MES pH 6.0, 50 mM NaCl, 2 mM DTT), bound to a 5 mL HiTrap SP column (GE) and eluted by a linear salt gradient. Fractions containing β-TrCP/Skp1 complex were concentrated and further purified by size exclusion chromatography (Superdex200 HiLoad 16/600, GE) (Supplementary Fig. 8). β-TrCP/Skp1 used in biochemical assays was buffer exchanged into 50 mM $NaH_2PO_4$, 200 mM NaCl, and 2 mM DTT during this final SEC step, concentrated to 100 µM, aliquoted, frozen in liquid nitrogen and stored at −80 °C. β-TrCP/Skp1 used in crystallization was buffer exchanged into 10 mM BTP pH 6.8, 200 mM NaCl, and 5 mM DTT during the final size exclusion step.

Full-length β-catenin (Myc/FLAG-tagged; Origene) was over-expressed in HEK293T cells for 48 h following transfection using FuGeneHD. In some instances, to increase yield, cells were treated with 5 µM Nedd8 E1 inhibitor (MLN4924) for 4 h prior to harvest. Cells were lysed in 50 mM Tris Cl (pH 7.5), 150 mM NaCl, 5% glycerol, 1% NP-40, protease and phosphatase inhibitors and bound to FLAG resin (Sigma), washed and eluted with 150 µg/ml 3× FLAG peptide. The eluted protein was concentrated, FLAG peptide was removed using spin column and analyzed (Supplementary Fig. 8).

Ubiquitin E1 (6xHis-Ube1), E2 (6xHis-TEV-Cdc34a), and Cul1-Rbx1 (6xHis-TEV-Cul1 and GST-TEV-Rbx1) were expressed in either *E. coli* or SF9 cells and purified as previously reported[35,36]. Briefly, proteins were purified on affinity columns followed by TEV cleavage, ion exchange chromatography and size exclusion chromatography (Supplementary Fig 8). Cul1-Rbx1 was neddylated in vitro using Nedd8-E1, Ubc12, and Nedd8 and subsequently purified to homogeneity (Supplementary Fig. 8).

**Fluorescence polarization (FP).** Compounds were incubated in a 20 µl reaction volume with β-TrCP/Skp1 (116 nM in binding enhancement assays and a concentration dilution series in peptide binding and cooperativity assays) and 5 nM BODIPY-TMR labeled β-catenin peptides (see figure legends for peptide details) in a black 384-well microplate (Perkin Elmer ProxiPlate), in a PBS buffer containing 0.01% Triton X-100 and 2 mM DTT with 2% final DMSO. Following 1-h incubation at room temperature, plates were read on an Envision plate reader (Perkin Elmer) for FP signal using a P-pol and S-pol 595/60 filter and a D555 single mirror. Binding and enhancement data were fit to a standard Hill equation ($Y = B_{min} + (B_{max} - B_{min})(X^\wedge h)/(X^\wedge h + EC_{50}^\wedge h)$) in Prism (GraphPad). Errors are standard error of the fit.

**Time-resolved fluorescence resonance energy transfer (TR-FRET).** Compounds were incubated in a 20 µl reaction volume with 300 pM biotinylated β-TrCP/Skp1 preincubated with 1.2 nM terbium-coupled streptavidin (Cisbio) and BODIPY-FL labeled β-catenin peptides (see figure legends for peptide details) in a white 384-well microplate (Perkin Elmer ProxiPlate), using a PBS buffer containing 0.01% Triton X-100 and 2 mM DTT with 2% final DMSO. In binding enhancement assays, the peptide concentrations were held constant at 107, 85, 10, 5, and 85 nM for the pSer33/Ser37, pS33/S37A, pS33/S37C, pS33/S37F, and S33E/S37A peptides, respectively. In peptide binding and cooperativity assays, the peptide concentrations were varied across the concentration range displayed on the *x*-axis. Following 1-h incubation, plates were read for TR-FRET signal after excitation at 320 nm and read at 520 and 620 nm using an EnVision plate reader (Perkin Elmer). TR-FRET signal was calculated using a 520/620 nm ratio. Binding and enhancement data were fit to a standard Hill equation ($Y = B_{min} + (B_{max} - B_{min})(X^\wedge h)/(X^\wedge h + EC_{50}^\wedge h)$) in Prism (GraphPad). Errors are standard error of the fit.

**High-throughput screen.** High-throughput screen activity was assessed with FP (as described above) and calculated using the percent change from the null control and standard deviation (σ) from the plate median. Hit thresholds were defined by two standard deviations from the plate median or 25% enhancement from the null control.

**Surface plasmon resonance (SPR).** SPR binding assays were performed on a GE Healthcare Biacore T200 using a Series B Biacore Sensor Chip SA. Biotinylated β-catenin pSer33/Ser37 peptide was immobilized at 27.6 RU. 333 nM β-TrCP/Skp1 and titrated compound were preincubated and flowed at a rate of 30 µl/min in 10 mM HEPES, 150 mM NaCl, 0.05% Surfactant P20, 1 mM TCEP, and 2% DMSO at pH 7.4 and 20 °C. Engagement of the ternary complex was assessed with a

contact and dissociation time of 120 s. Equilibrium binding levels were assessed using Biacore T200 Evaluation Software (GE) and these data were fit to a standard Hill equation ($Y = B_{min} + (B_{max} - B_{min})(X^\wedge h)/(X^\wedge h + EC_{50}^\wedge h)$) in Prism (GraphPad). Errors are standard error of the fit.

**Crystallization and structure determination.** Crystals were obtained by hanging drop vapor diffusion. A solution of 12 mg/mL β-TrCP/Skp1 in 10 mM BTP, 200 mM NaCl and 5 mM DTT containing a two-fold molar excess pSer33/Ser37 β-catenin (Ac-CDRKAAVSHWQQQSYLDpSGIHSGATTTAPSLSG) peptide were mixed one-to-one with a well solution of 0.1 M HEPES pH 7, 10% PEG 6000 and incubated over that well solution at 20 °C. Three to four crystals from this condition were pulverized in 100 µL of well solution using the Hampton Research Seed Bead kit and diluted 1:1000 with well solution. Subsequent crystals were obtained by adding 0.2 µL of this diluted seed stock to a 2 µL drop containing 1 µL of 8.2 mg/mL β-TrCP/Skp1 and a two-fold molar excess of either pSer33/Ser37 β-catenin peptide (Ac-CDRKAAVSHWQQQSYLDpSGIHSGATTTAPSLSG; pSer33/Ser37, NRX-1933, NRX-2663, and NRX-2776 containing crystals) or pSer33/S37A β-catenin peptide (Ac-CDRKAAVSHWQQQSYLD(pS)GIHAGATTTAPSLSG; NRX-103094 containing crystal), and 1 µL of a well solution containing 8% PEG4000, 100 mM BTP pH 5.5. After growth, crystals were transferred to a 2 µL drop of soaking solution containing 10% PEG 4 K, 0.1 M HEPES pH 7.1, 2 mM DTT, 400 µM β-catenin peptide, 10% DMSO and 1–10 mM of compound. These drops were suspended over a well of 10% PEG4000, 0.1 M HEPES pH 7.1 overnight. Crystals were cryoprotected in soaking solution supplemented with 18% ethylene glycol and frozen in liquid nitrogen.

Data for the pSer33/Ser37, NRX-2663, NRX-1933, and NRX-103094 containing crystals were collected at the Canadian lightsource on beamline 08ID at 100 K using and a wavelength of 0.9795 Å. Data for the NRX-2776 containing crystal were collected at the Advanced Photon Source on beamline 22ID at 100 K using and a wavelength of 1 Å. Data were processed using XDS and scaled in XSCALE, using CC1/2 as the resolution cutoff criteria[37]. For the highest resolution NRX-2776 containing structure, the coordinates of β-TrCP and Skp1 from PDB ID 1P22[30] were used as search ensembles in PHASER[38] to obtain an initial solution for this structure by molecular replacement. The final model was built through iterative rounds of refinement in Phenix[39,40] and Coot[41]. Ligand restraints were generated in Maestro and Phenix Elbow[42,43]. The resulting model for the NRX-2776 containing complex was used as the starting point and refinement constraints for refinement of the other structures (Supplementary Fig. 9). Model quality was determined by Molprobity[44]. The final models contained 95.5/4.5/0.0 (NRX-2776), 95.5/4.3/0.2 (NRX-2663), 95.5/4.3/0.2 (NRX-103094), 95.7/4.4/0.0 (NRX-1933), and 95.5/4.5/0.0 (pSer33/Ser37) percent Ramachandran favored/allowed/outliers. Figures were made using PyMol[45].

**pKa determination.** The pKa values were determined using the fast UV (spectrometric) technique. Samples were titrated in triplicate under methanol:water co-solvent conditions (with MeOH:water mixing ratios varied from 1:1 to 1:4) in the pH range 2.0–12.0 at concentrations of 15 to 30 µM and at a temperature of 25 °C. Aqueous pKa values were determined by Yasuda-Shedlovsky extrapolation from the individual determined results.

**Ligand docking.** The X-ray crystal structure of NRX-103094 was used in the docking of the various analogs. The structure was prepared with the protein preparation procedure in Maestro. All hydrogens were added, and charge states defined using PROTKA procedure, follow by h-bond optimization and constraint relaxation to 0.2 rmsd tolerance. The resulting structure was used to generate docking grids and perform docking studies using GLIDE. The GLIDE extra precision scoring function (XP) was used to obtain suitable docking poses, which were subsequently refined with Macromodel embrace procedure using default options. All the modeling was done with Maestro 10.3 Release 2015-2 (Schrodinger LLC).

**In vitro β-catenin ubiquitylation.** Four micromolar of fluorescently-labeled β-catenin peptide (residues 17–60) were ubiquitylated in a buffer containing 25 mM Tris Cl (pH 7.5), 50 mM NaCl, 5 mM $MgCl_2$, 2 mM ATP, and 1 mM DTT at room temperature in the presence of 62.5 µM Ubiquitin, 125 nM Ube1, 1.75 µM Cdc34A, and 100 nM neddylated $SCF^{\beta-TrCP}$ in the presence or absence of compound (as indicated). Typically, individual reaction mixes with either Ubiquitin, Ube1 and Cdc34A or β-catenin, $SCF^{\beta-TrCP}$ and compound were preincubated and subsequently mixed to initiate β-catenin ubiquitylation. Reactions were quenched by SDS loading buffer, resolved by SDS-PAGE and imaged for fluorescence. For full length β-catenin protein ubiquitylation 400 nM β-catenin and 400 nM $SCF^{\beta-TrCP}$ were used instead.

**Cellular pulldown experiments.** Lysate pulldown with recombinant β-catenin: HEK293T cells were lysed by freeze thaw in lysis buffer (1× PBS, 2 mM DTT, 0.01% Triton X-100, proteases inhibitors (Sigma, P2714 and 11836170001), phosphatase inhibitors (Sigma, P5726 and P0044), 5 nM Calyculin A (Sigma, C5552)) at 85 µL of buffer to 1 × 10^6 cells. Lysates were clarified by centrifugation at 20,000 × *g* for 20 min at 4 °C and the supernatant was collected for subsequent use. Purified Myc/FLAG-tagged wild-type β-catenin protein at 2 µM was pretreated

with 1 μM GSK3β (EMD Millipore, 14-306), 50 nM Axin (Enzo Life Sciences, ENZ-PRT130-0020), 200 nM CK1α (Life Technologies, PV3850), and 2 mM ATP at 30 °C for 60 min. This phosphorylated β-catenin, unphosphorylated WT β-catenin, or S33E/S37A β-catenin were each diluted into lysate to 100 nM, at the indicated compound concentrations, at a final DMSO concentration of 1% DMSO. Thirty-five microliters of these mixtures were incubated over 10 μL of anti-c-myc magnetic beads (Pierce, Catalog #PI88842), with rotation, at room temperature for 2 h. Beads were then pelleted, washed twice with 200 μL cold wash buffer (1×PBS, 2 mM DTT, 0.01% TX-100, 1% DMSO) containing compound at the same incubation concentration. Washed beads were resuspended in 30 μL SDS loading dye and analyzed by western analysis.

Pull down from cells treated with compound: HEK293T cells were plated in 10 cm dishes at a density of $1.2 \times 10^6$ cells per plate. Cells were transfected 24 h later with 2.5 μg of a WT or S33E/S37A Myc/FLAG-tagged β-catenin expression construct using the transfection reagent FuGeneHD. Forty-eight hours post transfection, cells were treated with a combination of 20 μM MG132 and NRX-252114 at the described concentrations. Cells were then liberated from 10 cm dishes using a cell scraper, pelleted, washed with PBS, resuspended in an equal volume of hypotonic lysis buffer (20 mM HEPES 7.5, 5 mM KCl, 1.5 mM MgCl$_2$, 2 mM DTT, 0.01% TX-100, protease and phosphatase inhibitors, 1% DMSO), and sheered by eight passages through 21 G and 25 G needles. Samples were then supplemented up to 150 mM NaCl and clarified by centrifugation. Thirty microliters of each lysate was incubated over 10 μL of anti-c-myc magnetic beads (Pierce, Catalog #PI88842) and incubated at room temp with rotation for 30 min. Beads were pelleted, washed twice with lysis buffer, and resuspended in SDS loading buffer. Samples were analyzed by western blot.

To monitor ubiquitylated β-catenin in lysates, treated cells (as indicated in figure) were lysed in lysis buffer (Cell Signaling Technologies, 9803) containing proteases inhibitors (Sigma, P2714 and 11836170001), phosphatase inhibitors (Sigma, P5726 and P0044), 5 nM Calyculin A (Sigma, C5552), 20 μM MG132 (Enzo Life Sciences, BML-PI102), 10 mM NEM (Sigma, 04259), 5 mM IAA (Iodoacetamide; Sigma, I1149), 2 mM OPT (1,10-Phenanthroline monohydrate; Sigma, 33510) and 50 μM PR-619 (LifeSensors, SI9619). Clarified lysates were resolved by SDS-PAGE followed by western analysis.

**S33E/S37A β-catenin stable cell line generation**. HEK293T cells were plated in a 10 cm dish in regular growth media (DMEM + 10% FBS) and allowed to adhere to the plate overnight. The following day, cells were transfected with S33E/S37 β-catenin using FuGeneHD transfection reagent. Transfected cells were then selected with media containing 700 μg/ml Geneticin. A pooled population of selected cells were used for all described experiments.

**β-catenin degradation with compound treatment**. HEK293T cells expressing S33E/S37A β-catenin were plated in six-well plates at a density of $9 \times 10^5$ cells/well. The following day, cells were incubated with compounds diluted in regular growth media to the appropriate concentration. The final concentration of DMSO for all cell treatments was kept at 0.5% or less. Cell treatments with the Nedd8 inhibitor (MLN4924) and the proteasome inhibitor (MG132) were performed at a final concentration of 5 μM and 10 μM respectively. Cycloheximide treatment was performed at a final concentration of 100 μg/ml.

**β-TrCP knockdown in S33E/S37A β-catenin expressing cell line**. HEK293T cells expressing S33E/S37A β-catenin were plated in six-well plates at a density of $5 \times 10^5$ cells/well. The following day, cells were transfected with a non-targeting siRNA or a combination of Dharmacon SMARTPool siRNAs targeting β-TrCP1 & β-TrCP2 using Dharmafect1 lipid reagent (Dharmacon/Horizon Discovery). Transfected cells were grown for 48 h before being re-plated in six-well plates at a density of $9 \times 10^5$ per well. The following day (72-h post siRNA transfection), cells were treated with either DMSO or NRX-252114 for 6 h. Harvested lysate was assessed for levels of FLAG-tagged S33E/S37A β-catenin by Western analysis.

**β-catenin shRNA knockdown**. TOV-112D cells were plated in six-well plates followed by infection with β-catenin shRNA lentivirus (TRC lentiviral constructs # NM_001904.3-2031s21c1, NM_001904.3-1541s21c1, and NM_001904.3-1309s21c1 from Sigma). Infected cells were then selected with blasticidin for three days before being used for subsequent experiments. Effects of β-catenin knockdown on cell proliferation were monitored using CelltiterGlo (Promega) reagent under standard conditions.

**Cellular ubiquitylation**. HEK293T or TOV-112D cells were treated with 20 μM MG132 for 4 h with or without 5 nM Calyculin A for 40 min. Cells were lysed in the presence of protease and phosphatase inhibitors, run on SDS-PAGE and analyzed by western analysis. The ubiquitylated β-catenin species were monitored using either anti-pS33 β-catenin antibody or anti-pS33/37 β-catenin antibody.

**Chemical syntheses**. Details of chemical syntheses can be found in the Supplementary Information.

**Peptides**. β-Catenin (17–48): Ac-CDRKAAVSHWQQQSYLDpSGIHSGATTT APSLSG

β-Catenin (17–60): Ac-CDRKAAVSHWQQQSYLDpSGIHSGATTTAPSLSGK GNPEEEDVDTS

IκBα: Ac-PRDGLKKERLLDDRHDpSGLDSMKDEEYEQMVKELQEIRLEPQEV

Wee1: Ac-CEEEEEEGSGHSTGEDpSAFQEPDSPLPPARSPTEPGPERRRSPG

Emi1: Ac-CSTNEIEALETSRLYEDpSGYSSFSLQSGLSEHEEGSLLEENFGDS

**Cell lines**. Cell lines were obtained from the American Type Culture Collection: HEK293T (ATCC- CRL-3216) TOV-112D (ATCC- CRL-11731).

**Antibodies**. Antibodies purchased from Cell Signaling Technologies: Anti-pS33/37 β-catenin #2009S (1:1000); Anti-pT41/S45 β-catenin #9565S (1:1000); Anti-β-catenin (C-ter) #9587S (1:2000); Anti-FLAG #2368 (1:2500); Anti-β-TrCP #4394 (1:1000). Anti-Actin was from Sigma, Catalog #A5441 (1:20,000), and Anti-pS33 β-catenin from GeneTex, Catalog #GTX50255 (1:1000)

**Reporting summary**. Further information on experimental design is available in the Nature Research Reporting Summary linked to this article.

## Data availability

A reporting summary for this article is available linked to this article. The source data underlying Figs. 1–6 are provided as a Source Data file. Additional data supporting the findings of this study are available from the corresponding author upon reasonable request. Structural coordinates for the complexes described has been deposited in the Protein Data Bank with the following PDB ID codes: 6M90 (NRX-2776), 6M91 (NRX-103094), 6M92 (NRX-2663), 6M93 (NRX-1933) and 6M94 (pSer33/Ser37).

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

## Acknowledgements
We thank Matt Clifton for assisting with the crystallographic data processing and discussion; Miklos Bekes for comments on the manuscript.

## Author contributions
K.R.S., K.B., C.P., and S.L.Y. with help from A.S. conducted all protein purification, biochemical, biophysical assays and compound profiling. K.R.S. did all the crystallography. J.T., Y.L., T.J.C., F.K., and M.C., with help from P.A.B. designed and synthesized the compounds. M.C. conducted the modeling and docking of ligand structures. K.R.S. and S.E.B. with the help of A.S. carried out the cell lysate experiments. K.B., S.E.B., and S.K. with help from N.F.B. and A.S. carried out the cellular experiments. Initial work was done in J.K. and M.R. labs. M.A.G, N.F.B., P.A.B. and A.S. supervised the project. K.R.S., J.T. and A.S. wrote the manuscript with input from all authors.

## Additional information

**Competing interests:** All authors with the exception of J.K., M.R. are either current or former employees and shareholders of Nurix Therapeutics, Inc. J.K. and M.R. are scientific founders of Nurix Therapeutics Inc.

