## [Peer Review File · Nature Communications]

REVIEWERS' COMMENTS:

Reviewer #1 (Remarks to the Author):

My comments remain the same. This work represents a nice proof concept of developing glue molecules to degrade oncogenic proteins. Unfortunately, compounds they generated most likely would not be useful to target beta-catenin. However, the work would certainly motivate people to use the similar strategy to degrade other oncogenic proteins. I would not be against its publication in Nature Communications.

Reviewer #2 (Remarks to the Author):

Overall the work presented here is of great quality and is an impactful and timely story for Nature readers. The authors report an interesting line of work regarding a proof-of-concept approach in developing "molecular glue" to potentiate the ubiquitination of β -catenin by its cognate E3 ubiquitin ligase, β -TrCP. The authors have presented a collection of beautiful data from the FP, TR-FRET and crystallographic work to the in vitro ubiquitination experiments presented in this manuscript. However, since S33 phosphorylation is dependent on S37 phosphorylation, this is not the most clinically relevant story. Can the authors maybe allude to any attempts that were made to identify "molecular glue" that promoted the binding of other S37 mutants? This could provide the reader with more context as to why the p33/S37A mutant was used in the proof-of-principle approaches presented. In addition to addressing this issue, are any of these enhancers also active against other mutants such as S37F or S37C? If this has not been evaluated, can the authors test this in an in vitro setting?

1. The last sentence of the abstract does not read comprehensively, "This prospective discovery of such 'molecular glue' provides a paradigm for the development of small molecule degraders of difficult-to-target proteins."

2. Even though the abbreviation for PPIs is defined in the abstract, it should be defined for the first time in the text of the manuscript as some readers will not look at the abstract.

3. References needed for: "These include immunosuppressants cyclosporine, FK506 and rapamycin as well as anti-cancer agents lenalidomide, pomalidomide and thalidomide (collectively referred as immunomodulatory drugs or IMiDs)."

4. In, "The use of PPI enhancer molecules as 'molecular glue' to potentiate substrate:ligase interactions is a therapeutically useful modality that might be employed to drug targets previously considered undruggable.", replacing might with "may" or "could" would make it more relevant to the present tense.

5. PROTACs remains undefined

6. "These challenges might be circumvented by discovering and rationally designing smaller molecular glue-like molecules that bind both the substrate and the ligase without the need for a linker to induce substrate degradation." This statement implies that the linker is the only "problem" with PROTACs, the authors should state that not only does molecular glue promote the interactions between two proteins, these molecules (i.e.; IMiDs and others listed) operate by presenting new potential points of interactions with the other protein binding partner that are otherwise not present.

7. "This work establishes a paradigm for rational design of small molecules to target oncogenic transcription factors by harnessing the ubiquitin proteasome system." It is unclear here why targeting transcription factors is the paradigm, this proof-of-concept approach just establishes that this molecular glue type strategy can be used to induce the interactions between a substrate protein and its cognate E3 ligase. This statement implies this is only applicable to oncogenic transcription factors, but that is not the case.

8. The authors state binding affinities throughout the results section, but fail to indicate the binding equilibrium equation(s) used to fit the data in both FP and TR-FRET. Did the authors make any assumptions when conducting binding experiments or did they use the classical quadratic equation defined for solving for concentration of bound species in a classical binding equilibrium. This should be the method used for calculating binding affinities in FP-based experiments described in the early results sections. Also, provide the software used to compute the data.

9. The authors are inconsistent with using the full-length name for amino acids versus the three-letter code.

10. The authors indicate that the pKa of NRX-1532 is 5.6. How did the authors calculate the pKa here? Is this a value from the literature, calculated using a particular software, or did the authors use potentiometric pH titrations to characterize this?

11. The authors show these enhancer molecules do not enhance the binding of other known β -TrCP substrates, such as Emi1, I κ B α and Wee1, but what about in a cellular context? How are the protein levels of these downstream targets affected by these molecules as compared to the changes observed in S33E/S37A β -catenin levels in Figure 6?

12. The authors state the levels of cooperativity observed for peptide binding with these molecules, however the authors do not describe how these cooperative values were calculated, what equation these data are fitted to and which software was used in the methods section. Except for NRX-252114 as described, "However, the full extent of cooperativity could not be measured in this assay, since the lower ligand concentration reaches the concentration of β -TrCP, resulting in a curve hill slope >1 ."

13. The authors fail to provide error for any of the EC50s described.

14. Subtitle, "Enhancers degrade S37A mutant β -Catenin in an engineered cellular system" suggests these molecules literally degrade β -Catenin. This should always be described as promoting the ubiquitination and subsequent degradation, as degradation is mediated by the 26S proteasome.

15. Cellular Ubiquitylation protocol should include lysis buffer conditions and product numbers for antibodies used for western blotting.

16. ^{13}C and ^{19}F NMR is missing for most molecules.

Manuscript title: Prospective discovery of small molecule enhancers of an E3 ligase-substrate interaction

Authors: Kyle R. Simonetta, Joshua Taygerly, Kathleen Boyle, Stephen E. Basham, Chris Padovani, Yan Lou, Thomas J. Cummins, Stephanie L. Yung, Szerenke Kiss von Soly, Frank Kayser, John Kuriyan, Michael Rape, Mario Cardozo, Mark A. Gallop, Neil F. Bence, Paul A. Barsanti and Anjanabha Saha

We thank the reviewers and the editorial team for their thoughtful feedback and critique. Below we included responses to the specific points raised by the reviewers.

Referee #1 (Remarks to the Author):

My comments remain the same. This work represents a nice proof concept of developing glue molecules to degrade oncogenic proteins. Unfortunately, compounds they generated most likely would not be useful to target beta-catenin. However, the work would certainly motivate people to use the similar strategy to degrade other oncogenic proteins. I would not be against its publication in Nature Communications.

Response: We thank the reviewer for their comments and wholeheartedly agree with their sentiment that a similar approach could be used to target hard to drug oncogenic proteins.

Referee #2 (Remarks to the Author):

Overall the work presented here is of great quality and is an impactful and timely story for Nature readers. The authors report an interesting line of work regarding a proof-of-concept approach in developing “molecular glue” to potentiate the ubiquitination of β -catenin by its cognate E3 ubiquitin ligase, β -TrCP. The authors have presented a collection of beautiful data from the FP, TR-FRET and crystallographic work to the in vitro ubiquitination experiments presented in this manuscript. However, since S33 phosphorylation is dependent on S37 phosphorylation, this is not the most clinically relevant story. Can the authors maybe allude to any attempts that were made to identify “molecular glue” that promoted the binding of other S37 mutants? This could provide the reader with more context as to why the p33/S37A mutant was used in the proof-of-principle approaches presented. In addition to addressing

this issue, are any of these enhancers also active against other mutants such as S37F or S37C? If this has not been evaluated, can the authors test this in an in vitro setting?

Response: We thank the reviewer for their comments. As part of the study, we evaluated S37F and S37C β -Catenin mutants in the *in vitro* peptide binding assay. Those results are summarized in Supplementary Table 2b. These measurements were done with the earlier compounds rather than the final compound since the series was optimized around S37A β -Catenin mutant.

Other points:

1. The last sentence of the abstract does not read comprehensively, "This prospective discovery of such 'molecular glue' provides a paradigm for the development of small molecule degraders of difficult-to-target proteins."

Response: We have appropriately modified the sentence.

2. Even though the abbreviation for PPIs is defined in the abstract, it should be defined for the first time in the text of the manuscript as some readers will not look at the abstract.

Response: The definition of PPI has been added to the main text.

3. References needed for: "These include immunosuppressants cyclosporine, FK506 and rapamycin as well as anti-cancer agents lenalidomide, pomalidomide and thalidomide (collectively referred as immunomodulatory drugs or IMiDs)."

Response: The above section is appropriately referenced in the revised version.

4. In, "The use of PPI enhancer molecules as 'molecular glue' to potentiate substrate:ligase interactions is a therapeutically useful modality that might be employed to drug targets previously considered undruggable.", replacing might with "may" or "could" would make it more relevant to the present tense.

Response: We have updated the text and used "could" instead.

5. PROTACs remains undefined

Response: We have defined PROTACs in the main text.

6. "These challenges might be circumvented by discovering and rationally designing smaller molecular glue-like molecules that bind both the substrate and the ligase without the need for a linker to induce substrate degradation." This statement implies that the linker is the only "problem" with PROTACs, the

authors should state that not only does molecular glue promote the interactions between two proteins, these molecules (i.e.; IMiDs and others listed) operate by presenting new potential points of interactions with the other protein binding partner that are otherwise not present.

Response: We have appropriately modified the sentence by calling out that molecular glues can promote interaction with either native- or neo-substrates.

7. "This work establishes a paradigm for rational design of small molecules to target oncogenic transcription factors by harnessing the ubiquitin proteasome system." It is unclear here why targeting transcription factors is the paradigm, this proof-of-concept approach just establishes that this molecular glue type strategy can be used to induce the interactions between a substrate protein and its cognate E3 ligase. This statement implies this is only applicable to oncogenic transcription factors, but that is not the case.

Response: We have appropriately modified the sentence and state that this is a general approach for degrading proteins, including oncogenic transcription factors (rather than only oncogenic factor). This is an important point and thank the reviewer for the suggestion.

8. The authors state binding affinities throughout the results section, but fail to indicate the binding equilibrium equation(s) used to fit the data in both FP and TR-FRET. Did the authors make any assumptions when conducting binding experiments or did they use the classical quadratic equation defined for solving for concentration of bound species in a classical binding equilibrium. This should be the method used for calculating binding affinities in FP-based experiments described in the early results sections. Also, provide the software used to compute the data.

Response: The equations used to fit the data and software used are described in the Methods section under FP and TR-FRET assays.

9. The authors are inconsistent with using the full-length name for amino acids versus the three-letter code.

Response: The amino acids are now referred by 3 letter code.

10. The authors indicate that the pKa of NRX-1532 is 5.6. How did the authors calculate the pKa here? Is this a value from the literature, calculated using a particular software, or did the authors use potentiometric pH titrations to characterize this?

Response: The pKa was determined experimentally and we have included the method in this revised version.

11. The authors show these enhancer molecules do not enhance the binding of other known β -TrCP substrates, such as Emi1, I κ B α and Wee1, but what about in a cellular context? How are the protein levels of these downstream targets affected by these molecules as compared to the changes observed in S33E/S37A β -catenin levels in Figure 6?

Response: In cellular context we have only tested WT β -Catenin and I κ B α and saw no changes. We have included a sentence to mention this in the revised version.

12. The authors state the levels of cooperativity observed for peptide binding with these molecules, however the authors do not describe how these cooperative values were calculated, what equation these data are fitted to and which software was used in the methods section. Except for NRX-252114 as described, "However, the full extent of cooperativity could not be measured in this assay, since the lower ligand concentration reaches the concentration of β -TrCP, resulting in a curve hill slope >1 ."

Response: The individual binding affinities of the peptide for β -TrCP in the absence and presence of saturating enhancer concentrations were determined by the FRET-based peptide binding titration method and fitted using the equations outlined in the methods. The cooperativities of the compounds were determined simply as the ratio of these two values.

13. The authors fail to provide error for any of the EC50s described.

Response: In the revised version, errors are noted for the enhancer values.

14. Subtitle, "Enhancers degrade S37A mutant β -Catenin in an engineered cellular system" suggests these molecules literally degrade β -Catenin. This should always be described as promoting the ubiquitination and subsequent degradation, as degradation is mediated by the 26S proteasome.

Response: This is a good point and we have appropriately edited the title.

15. Cellular Ubiquitylation protocol should include lysis buffer conditions and product numbers for antibodies used for western blotting.

Response: In the revised version, we have added the lysis protocol in the methods section.

16. ^{13}C and ^{19}F NMR is missing for most molecules.

Response: All final compounds have been characterized by ^{13}C and ^{19}F NMR, and the tabulated data is included in the supplementary information.